

# Wintertime Production and Storage of Methane in Thermokarst Ponds of Subarctic Norway

Anfisa Pismeniuk[1,2], Peter Dörsch[2,3], Mats R. Ippach[1,2], Clarissa Willmes[1], Sunniva Sheffield[4], Norbert Pirk[1,2], Sebastian Westermann[1,2]

[1]Department of Geosciences, University of Oslo, Oslo, 0371, Norway
[2]Centre for Biogeochemistry in the Anthropocene, University of Oslo, Oslo, 0371, Norway
[3]Faculty of Environmental Sciences and Natural Resource Management, Norwegian University of Life Sciences (NMBU), Ås, 1433, Norway
[4]Department of Chemistry, University of Oslo, Oslo, 0371, Norway

*Correspondence to*: Anfisa Pismeniuk (anfisa.pismeniuk@geo.uio.no)

**Abstract.** The ongoing climate change in permafrost areas can trigger abrupt thaw processes, leading to the formation of thermokarst lakes and ponds. These water bodies, especially in organic-rich areas, are recognized as strong methane emitters during the ice-free periods and have the potential to accumulate high amounts of methane in and under the ice, which can be released during the ice melt. We estimated wintertime $CH_4$ storage and daily bottom flux in nine shallow ponds within two
permafrost peatlands in Northern Norway, Iškoras and Áidejávri, during the 2023–2024 ice cover season. The wintertime $CH_4$ storage ranged from 0.6 to 24 g $CH_4$-C m$^{-2}$ and contributed up to 40 % of the annual $CH_4$ budget at the Iškoras site. The heterogeneity of the $CH_4$ wintertime accumulation is related to pond depth, differences in vegetation, and the thermokarst pond formation age. The latter has been investigated using a space-for-time substitution approach along chronosequences of thermokarst formation spanning more than 70 years. The winter $CH_4$ bottom flux increased from 3 mg $CH_4$-C m$^{-2}$ d$^{-1}$ in two-
year-old pond to 107 mg $CH_4$-C m$^{-2}$ d$^{-1}$ in a pond formed between 30 and 60 years ago. Ponds that formed more than 70 years ago and are currently experiencing sedge regrowth exhibited a high $CH_4$ bottom flux of 60 mg $CH_4$-C m$^{-2}$ d$^{-1}$, while older ponds dominated by Sphagnum mosses showed 4 to 10 times lower $CH_4$ bottom fluxes.

## 1 Introduction

Lakes and ponds are key sources of methane ($CH_4$) at high northern latitudes (Wik et al., 2016), particularly in ice-rich
permafrost regions where abrupt thaw processes lead to the formation of thermokarst waterbodies (Turetsky et al., 2020). Thermokarst lakes and ponds are recognized as strong methane sources (Heslop et al., 2015; Vonk et al., 2015; Kuhn et al., 2021, Knutson et al., 2025). Recent synthesis studies have summarized existing data on $CH_4$ fluxes from waterbodies in boreal and Arctic ecosystems (Wik et al., 2016; Denfeld et al., 2018; Kuhn et al., 2021). However, most of this data represents ice-free periods, while studies addressing seasonal variations remain limited. Northern lakes are ice-covered for
roughly 60 % of the year underscoring the importance of estimating wintertime $CH_4$ storage and bottom flux during frozen periods (Walter Anthony et al., 2010, Wik et al., 2011; Boereboom et al., 2012; Greene et al., 2014, Langer et al., 2015). Ice-



covered lakes and ponds often experience anoxic conditions which lead to significant $CH_4$ accumulation both under and in the ice. This methane is released upon ice-off from the oversaturated water column and from ice bubble storage (Greene et al., 2014; Sepulveda-Jauregui et al., 2015; Vonk et al., 2015). It remains unclear to what extent the emission of $CH_4$

accumulated during winter contributes to the annual $CH_4$ budget of arctic landscapes. Several studies have reported high emissions during ice melt (Phelps et al., 1998, Karlsson et al., 2012; Jammet et al., 2015), but quantification of wintertime $CH_4$ production in northern lakes and ponds is scarce (Boereboom et al., 2012; Langer et al., 2015) and may result in underestimation of annual $CH_4$ emissions, especially for peatland thermokarst lakes (Matveev et al., 2019).

Northern peatlands cover approximately $3.7 \pm 0.5$ million km² and store around $415 \pm 150$ Pg of carbon (Hugelius et al.,

2020). Given this large carbon pool, decomposition of peat may trigger significant additional greenhouse gas (GHG) emissions, potentially amplifying global warming. The sporadic permafrost zone outside the mountain regions of Fennoscandia is largely represented by peat plateaus, which are currently undergoing significant degradation due to thermokarst processes and may completely disappear from most of these areas in the coming decades (Borge et al., 2017). The intensification of abrupt thaw is leading to the expansion of small (< 1000 m² in surface area) and shallow (< 2 m in

depth) thermokarst lakes and ponds. Despite their significance as active $CH_4$ sources, these small water bodies are often overlooked in the global assessments, although high $CH_4$ emissions have been reported in Canada, Northern Sweden, and Russia (Matveev et al., 2016; Kuhn et al., 2018; Burke et al., 2019; Serikova et al., 2019).

In this study, we assessed wintertime $CH_4$ production and storage in nine ponds across two permafrost peatland sites within the sporadic permafrost zone of Finnmark, Northern Norway. We estimated $CH_4$ storage under and in the ice during the

2023-2024 winter season. Our specific objectives were to (1) quantify $CH_4$ winter bottom fluxes in various ponds across two permafrost peatland sites in Finnmark, (2) assess the contribution of wintertime $CH_4$ storage to the annual $CH_4$ budget at one of the sites, (3) identify the main factors causing differences in $CH_4$ winter bottom fluxes, and (4) explore the relationship between the age of thermokarst pond formation and $CH_4$ winter bottom fluxes.

## 2 Study area

We investigated nine ponds located within two peat plateau complexes in the sporadic permafrost zone of Finnmark, Northern Norway (Fig. 1a): Iškoras (69°20'26.4"N; 25°17'42.2"E; 380 m a.s.l.) and Áidejávri (68°45'14"N; 23°19'01"E, 398 m a.s.l.). These peatlands have experienced thawing for decades, and the thermokarst pond formation has accelerated significantly in the last 20 years (Borge et al., 2017). Using a chronosequence approach, the formation age of each thermokarst pond was determined using historical aerial images from the Norwegian Mapping Authority (Norgeibilder.no,

2025) dating back to 1955 for Iškoras and 1958 for Áidejávri, as well as drone survey images obtained between 2015 and 2023 for both sites (method as described in Martin et al., 2021). The thermokarst pond formation age was identified as the time of permafrost collapse, which typically coincided with water accumulation; however, there can be exceptions with the accumulation of water starting several years after collapse, as observed in the case of pond A4 (Fig. S1). The year 2023





served as the baseline for age estimations. While some thermokarst ponds are currently undergoing a transition to terrestrial

wetland ecosystems through sedge and/or peat succession, recently formed ponds continue to expand due to thawing and

collapse of peat plateau edges.

**Figure 1. Study area: (a) map showing permafrost distribution in Northern Scandinavia and surrounded territories with the location of two studied peat plateaus (adapted from Obu et al., 2019); (b) studied ponds in Áidejávri (A1-A8); (c) studied pond Isk**

**in Iškoras with two sampling locations (Isk-1 and Isk-2). The colour of the ponds represents the thermokarst pond formation age (Sect. 2). The orthophotos used in (b) and (c) were obtained from a drone survey conducted in 2023 and processed as described in Martin et al., 2021.**

Both peat plateau complexes are located within the continental subarctic climate zone, with a mean annual air temperature of

-2°C at Áidejávri (meteorological station Sihccajavri) and -1.5°C at Iškoras (meteorological station Cuovddatmohkki) for the

period 1980-2024 (https://seklima.met.no). The mean air temperature measured at 2 m in Iškoras over the past three years,

recorded by the eddy covariance tower at the Iškoras site (Pirk et al., 2024) was 0.15°C. The ice growth season at Iškoras in

2023 began between October 1 and 2, coinciding with the first substantial drop in air temperature (Fig. S2). In Áidejávri, the

onset of ice growth occurred between October 6th and 13th as indicated by temperature data from the meteorological station

Sihccajavri and Sentinel-2 satellite imagery (Fig. S3). During the sampling in March 2024, the snow thickness varied from




0.2 to 0.8 m (Table 1). Ponds less than 0.5 m deep (Table 1) were frozen to the bottom, including the bottom peat layers
(ponds A5 and A6), which correspond to the former surface layers of peat plateaus that became submerged due to
thermokarst formation. The ice cover remained until mid-May 2024 and disappeared completely by the end of May (Fig.
S4). During the fieldwork in October 2024, the ponds at both sites were ice-covered since the 4th of October.

At Áidejávri, we focused on eight shallow ponds (ranging from 0.3 to 1.5 m in depth) located in the northern and eastern
parts of the large peat plateau complex (Fig. 1b, Table 1). In the main study area, we studied six thermokarst ponds (A1–A6)
formed within the last 80 years. The ponds were selected and named in chronological order, the lowest number representing
the oldest pond. All ponds covered an area < 500 m². Pond A1, the oldest in the study area, and pond A3, formed over the
last two decades, are rapidly overgrowing with sedges. Ponds A2 and A4-6 are currently expanding and are to some degree
hydrologically connected, but initially formed as separate water bodies. The ponds A2, A4-6 share similar vegetation and
location conditions but differ in their thermokarst pond formation ages (Table 1). For comparison, we included two reference
ponds (A7 and A8) located to the north and east of the main study area at Áidejávri (Fig. 1b). Pond A7, situated in the
eastern part and being the oldest studied, has retained its boundaries since 1958 and is currently undergoing slow peat
formation (Table 1). Pond A8, which has also been stable since 1958, differs in shape from typical thermokarst ponds in
permafrost peatland environments and was therefore classified as non-thermokarst.

At Iškoras, the majority of thermokarst ponds is located along the southern edge of the plateau (Fig. 1c). Here, we sampled
two locations in the oldest Isk pond (Fig. 1c). The sampling locations represent different pond formation ages, with the
oldest part (Isk-1) formed before 1955 and the younger part collapsed between 1955 and 2003 (Isk-2). The pond depth was
estimated to be 0.6 m at both locations (Table 1), but the depth determination was challenging due to plant growth and new
peat formation within the pond. Despite the ongoing retreat of the surrounding peat plateau edges, the pond borders appear
relatively stable. The pond is undergoing Sphagnum colonization, especially on its eastern shore, indicating that it is
approaching the final stage of thermokarst pond development. More detailed information about the ponds is given in Table 1.
The water temperature in Isk pond at Iškoras was measured at two depths (0.4 m and 0.6 m) using automatic data loggers
(Fig. S5), indicating that the pond is well mixed during the ice-free period. However, temperature differences of up to 5°C
between the two depths were observed during the initial ice formation and ice melt periods. In winter, the temperature varied
due to top-down ice formation which reached the upper logger, but temperature differences did not exceed 2°C.



**Table 1. Characteristics of the studied ponds. The thermokarst pond formation age was determined as the period of peat plateau collapse based on historical aerial images. Numbers in parentheses represent the thermokarst pond formation age used for comparisons and figures in this study (the year 2023 was the baseline for age estimations). Snow and ice thicknesses were measured during the field campaign in March 2024. Total ice thickness included the frozen bottom peat layer in ponds A5 and A6, which corresponds to the former surface of the peat plateau that has subsided due to thermokarst.**

| Pond name | Area, $m^2$ | Depth, m | Stage | Thermokarst pond formation age | Total ice thickness, m | Snow thickness, m |
|---|---|---|---|---|---|---|
| **Áidejávri, main study area** | | | | | | |
| A1 | 8 | 1.3 | overgrowing with sedges | > 65 (70) | 0.49 | 0.2 |
| A2 | 88 | 1.5 | expanding | 20-65 (40) | 0.46 | 0.5 |
| A3 | 53 | 1.1 | overgrowing with sedges | 10-20 (15) | 0.53 | 0.4 |
| A4 | 112 | 0.8 | expanding | 3-10 (7) | 0.58 | 0.3 |
| A5 | 84 | 0.4 | expanding | 3-7 (5) | 0.56 | 0.8 |
| A6 | 27 | 0.4 | expanding | 1-3 (2) | 0.53 | 0.8 |
| **Áidejávri, reference ponds** | | | | | | |
| A7 | 1386 | 0.5 | stable, new peat formation | > 65 (100) | 0.58 | 0.3 |
| A8 | 2577 | 0.8 | non-thermokarst pond | | 0.45 | 0.2 |
| **Iškoras, Isk pond** | | | | | | |
| Isk-1 | 538 | 0.6 | stable, new peat formation | > 68 (70) | 0.50 | 0.6 |
| Isk-2 | 538 | 0.6 | stable, new peat formation | 20-68 (44) | 0.55 | 0.5 |

## 3 Methods

### 3.1. Field Sampling

Fieldwork was carried out in 2024 during March 11–17, September 10–13, and October 4–5. During the winter sampling in March 2024, ice cores and below-ice pond water samples were collected in the center area of each pond. At each sampling location, after measuring the snow thickness, we cleaned the ice surface and drilled through the ice using an ice auger. If the pond was sufficiently deep, water was sampled from both near the bottom and directly below the ice table using a custom-made sampler consisting of a 120 mL serum bottle on a rotating arm. The sampler was lowered to a specific depth with the bottle neck pointing downwards, before turning the bottle upwards to fill it with water. Immediately after bringing the water samples to the surface, dissolved gases were extracted on-site using the acidified headspace method (Åberg and Wallin, 2014). For this, we collected 30 mL of water with a disposable syringe equipped with a 3-way valve and created a 20 mL headspace with ambient air before adding 0.6 mL of 3 % HCl. After shaking for five minutes, the headspace gas was transferred to Helium (He) washed and evacuated 12 mL septum vials (Chromacol). Ambient air was collected at each sampling point to correct for background concentrations for gas extraction. After taking dissolved gas samples, another 50



mL of water from the serum bottles was transferred into 50 mL Falcon tubes for dissolved organic carbon (DOC) analysis. These samples were filtered through a 0.45 μm sterile syringe filter with an RC membrane (VWR International) on the same day and subsequently stored at 4°C before being analyzed. Water temperature and pH were measured in the remaining water sample.

After processing the water samples, we extracted ice columns close to the sampling hole using an ice auger. The ice core was
subsampled horizontally according to visible differences in texture, shipped frozen to a storage in a cold container (-5°C) at the University of Oslo. In September 2024, we repeated the sampling of dissolved gases and water at the same locations using the procedure described above, except that water was collected from a depth of 0.1 m directly from the pond for dissolved gas measurements. Water for DOC measurements was sampled from the same location and depth, pH and temperature were recorded. In October 2024, additional water samples were extracted for dissolved gases from selected
ponds (A1, A2, A4, A5, Isk) both from below the first ice formed and the deeper parts of the pond, again using the water sampler and the acidified headspace method described above.

### 3.2 Ice sample preparation

Ice samples were cleaned and cut in the cold container at -5°C at the University of Oslo. The bubble structure was visually described, partly following the classification of Boereboom et al. (2012). Samples were characterized as "Superimposed ice",
"Clear ice", "Spherical and nut-shaped bubbles", "Elongated bubbles", "Mixed bubbles", "Methane ebullition bubbles" and "Frozen peat". Superimposed ice was found on a top of the ice cores, identified by its texture, brownish color, higher impurities content and high DOC content compared to the ice layers below (Manispurov et al., 2015). Methane ebullition bubbles were identified as relatively large (1-2 cm in diameter), flat bubbles near the surface layers. In the deeper layers, ebullition bubbles were categorized as mixed bubbles.

When enough sample material was available, the ice monoliths were divided into three subsamples. Each subsample was placed into a 1050 mL glass jar and sealed with an airtight lid equipped with a sampling septum. The jars were flushed with He using an automated manifold and a vacuum pump, and after releasing He-overpressure, the ice was left to melt at room temperature (+23°C) overnight. The bottles were shaken vertically at 120 rpm for 1 hour to equilibrate gases between the sample and the headspace. Immediately following shaking, the jar headspace was analyzed for $CO_2$ and $CH_4$ using a gas
chromatograph (GC).

Thereafter, the meltwater was collected to measure the ice water equivalent, pH and DOC content, using the same filtration procedure as for the pond water samples (Sect. 3.1). If the bottom samples contained plant material or peat, they were dried in an oven at +40°C to estimate dry weight and liquid volume. A bulk density of the peat of 0.2 g cm$^{-3}$ was calculated from the dry weight of ca. 100 g of field-wet sample of known volume after freeze-drying for at least 72 h.





### 3.3 Analysis of gas concentrations

Dissolved gases from both the ponds and the ice monoliths were analyzed at the Norwegian University of Life Sciences (Ås, Norway) using a gas chromatograph (GC; Model 7890A, Agilent, Santa Clara, CA, USA) equipped with an autosampler (GC-Pal, CTC, Switzerland). Approximately 2 mL of headspace gas was sampled by a hypodermic needle connected to a peristaltic pump (Gilson Minipuls 3) and admitted to two heated 250 µL sampling loops loading the analyte on two separation columns: a 20 m wide-bore (0.53 mm) Poraplot Q column for the separation of $CH_4$, $CO_2$, and $N_2O$ from bulk gases, and a 60 m wide-bore Molsieve 5Å PLOT column for separating Ar, $O_2$, and $N_2$. Calibration and conversion of peak areas to ppm were performed using dry bottles with standard gas mixtures (AGA, Norway). The precision of the GC measurements was within 1 % determined by repeated analyses of certified gas standards.

Dissolved concentrations at in situ water temperature were calculated from measured headspace concentrations using temperature-corrected solubility constants (Wilhelm et al., 1977) considering the temperature and volume of the sample at extraction. Dissolved $CO_2$ concentrations were back calculated to in situ pH using bicarbonate equilibrium constants (Appelo and Postma, 1993). Dissolved oxygen ($O_2$) concentrations were analyzed for the samples collected in September 2024. To assess the oxygen conditions in the ponds in September 2024, we determined the $O_2$ saturation in the water relative to atmospheric equilibrium. We classified the oxygen conditions as oxic when the saturation exceeded 30 %, as hypoxic when the saturation was below 30 %, and as anoxic when saturation values were below 1 %. We estimated a high absolute error of 20 % in oxygen saturation measurements due to significant differences observed between measured values in replicate samples. Dissolved oxygen concentrations below ice were not measured.

### 3.4 pH and Dissolved Organic Carbon (DOC) analysis

The pH was measured using a HANNA instruments pH meter HI9124 with a HI1230B pH electrode, which was 3-point calibrated using Cetripur ® buffer solutions from Supelco ® (buffer solutions pH 4.01, 7, 10).

The Dissolved Organic Carbon content (DOC) was analyzed using the Total Organic Carbon Analyser (TOCV, Shimadzu, Japan) coupled to an autosampler (ASI-V) using combustion and near infrared detection of $CO_2$ after removing carbonates by HCl.

### 3.5 Winter methane bottom flux

In this study, we determine the winter methane storage ($CH_4$) as the sum of methane stored in the ice and in the unfrozen water below the ice. To derive the "winter methane bottom flux" (i.e. the methane flux at the sediment-water interface), we subtract the $CH_4$ storage prior to freezing (see below) and divide by the time interval between the start of ice formation and the day of sampling in March 2024. The $CH_4$ storage in the ice was determined by combining the concentrations of the individual ice layers, excluding superimposed ice (Sect. 3.2) which was in contact with the atmosphere. Frozen peat was considered separately; its volumetric $CH_4$ content was calculated from the concentration in the water phase, corrected by the



volume of the peat using the bulk density (Sect. 3.2). We estimated the uncertainty for individual ice layers using the standard deviation of all available subsamples. In cases where the number of subsamples was insufficient, we assigned an uncertainty based on the average relative error observed in comparable ice layers (Sect. 3.2) from similar ponds. For the frozen peat samples, we calculated the uncertainty by applying the average relative error derived from the deepest ice layers

in other ponds. The uncertainty of each layer thickness was estimated during the sample preparation to be around 0.02 m.

We also estimated the potential contribution of superimposed ice, for which it is unclear whether the contained $CH_4$ is derived from the air (in case of frozen melt- or rainwater) or from pond water pressed upwards through cracks.  In eight of the ice cores, the superimposed ice contributed less than 1% of the total $CH_4$ storage, while it was 5 % for pond A3. In these cases, the uncertainty due to the superimposed ice was considered negligible compared to the uncertainty of the $CH_4$

concentrations in the individual ice layers. However, in pond A6, superimposed ice comprised half of the ice column, contributing 50 % to the total $CH_4$ storage. For this pond, we therefore assumed a 50 % uncertainty in the ice storage.

The $CH_4$ storage in the water column under the ice was estimated for ponds that had not frozen completely to the bottom. We assumed a 5 % uncertainty for the $CH_4$ measured by the field headspace method, based on existing uncertainty estimates of the method (Åberg and Wallin, 2014; Koschorreck et al., 2021). The presence of plants or loose peat layers on the pond

bottoms created challenges for determining whether the probe reached the true bottom of the pond, and we therefore assumed a depth uncertainty of ± 0.1 m. For the pond at Iškoras, disturbances caused by the drilling made it impossible to determine whether it was frozen to the bottom or whether a 0.05 to 0.1 m thick water layer remained below the ice. Furthermore, it was not possible to collect dissolved $CH_4$ samples from the stirred-up water. As even a thin water layer can contain large amounts of $CH_4$, we report two confining $CH_4$ fluxes for the Isk pond: one assuming that the pond had frozen

completely to the bottom, and a second flux estimate assuming a water layer of 0.05 m (Isk-2) and 0.1 m (Isk-1) with $CH_4$ concentration corresponding to the average $CH_4$ partial pressures from the Áidejávri ponds.

As we do not have data for the water column $CH_4$ storage prior to freezing in October 2023, we use two confining estimates. First, we calculate the average of the $CH_4$ concentrations measured in September 2024, with solubility constants corrected for a temperature of 0°C, at which the actual pond freezing occurs. Secondly, we use the $CH_4$ concentrations in the

uppermost winter ice layers which might reflect the gas concentrations in the water prior to freezing. As both estimates are associated with considerable uncertainty, we use the average of both estimate and assign the range to the maximum and minimum values as uncertainty (Sect. 4.3). For the pond depth, we estimated uncertainties of ± 0.1 m for the Áidejávri ponds, and ± 0.2 m for Iškoras. We emphasize that the $CH_4$ storage prior to freezing is generally small compared to the combined ice and water storage during winter, so that it does not contribute strongly to the overall uncertainty of the winter

methane bottom fluxes.

The accumulation period was estimated using meteorological data and Sentinel-2 satellite imagery available for October 2023 (Sect. 2 and Fig. S2-S3). Initial ice formation on both peat plateaus started between October 1 and 13, 2023, while sampling was conducted from March 11 to 17, 2024. For calculation purposes, we set the accumulation period from October 7, 2023 to March 14, 2024, and considered a temporal error of ± 10 days.





To compute the uncertainty of our winter $CH_4$ bottom flux estimates, we use Gaussian error propagation taking into account all the uncertainties of the individual terms ($CH_4$ ice and water storage, ice and water depth/thickness, freezing period, etc.). Spearman's rank correlation coefficients ($r_s$) were calculated to assess relationships between biogeochemical pond parameters, estimated fluxes, and environmental factors. The strength of the correlation was characterized as weak ($r_s$=0.20-0.39), moderate ($r_s$=0.40-0.59), strong ($r_s$=0.60-0.79) and very strong ($r_s$=0.80-1). A p-value smaller than 0.05 was

considered a statistically significant correlation.

## 4 Results

### 4.1 Seasonal variations of pH, DOC, and dissolved $CH_4$ and $CO_2$ in water

All studied ponds were acidic, with pH levels ranging from 4 to 6.4 (Table 2). The highest pH (6.4) was measured in the non-thermokarst pond A8. Apart for the latter, the pH increased from September to March for all ponds not frozen to the

bottom.

**Table 2. Water column pH and Dissolved Organic Carbon (DOC) concentration in studied ponds sampled in March and September 2024. Ponds A5-A7 were frozen to the bottom in March 2024.**

| Site | | | | | Áidejávri | | | | Iškoras |
|---|---|---|---|---|---|---|---|---|---|
| | A1 | A2 | A3 | A4 | A5 | A6 | A7 | A8 | Isk |
| **pH** | | | | | | | | | |
| September | 4 | 4.4 | 5.5 | 4.2 | 4.1 | 4 | 4.7 | 6.4 | 4.2 |
| March | 4.8 | 4.9 | 5.9 | 5 | frozen | frozen | frozen | 6.2 | 5 |
| **DOC, mg L⁻¹** | | | | | | | | | |
| September | 95 | 85 | 75 | 90 | 105 | 105 | 36 | 44 | 42 |
| March | 58 | 78 | 70 | 95 | frozen | frozen | frozen | 36 | 50 |

DOC concentrations varied from 36 mg L⁻¹ in the non-thermokarst pond A8, sampled in March, to 105 mg L⁻¹ in the young,

shallow ponds A5 and A6, sampled in September (Table 2). In contrast to pH, DOC concentrations showed no clear seasonal pattern. The largest seasonal difference was observed in pond A1, which is overgrowing with sedges (Table 2). A strongly negative correlation (p < 0.05) was observed between pond size and DOC concentration in September. Concentrations were low (< 50 mg L⁻¹) in the ponds larger than 500 m² A7 and A8 in Áidejávri, and Isk in Iškoras. Although September DOC concentrations showed no distinct pattern with pond age, there was a moderate negative correlation (p < 0.05) with pond age

in the main study area (A1-A6) in Áidejávri. DOC concentrations in the overgrowing ponds A1 and A3 ranged from 58 to 95 mg L⁻¹. DOC concentrations in A2, A4, A5, and A6 averaged at 91 mg L⁻¹ and correlated negatively with ponds age, increasing from 78 mg L⁻¹ in the oldest pond A2, to a maximum of 105 mg L⁻¹ in the recently formed ponds A5 and A6.



Large ($> 500 \text{ m}^2$) ponds (A7, 8 and Isk) were saturated with oxygen in September (Fig. S6), while the smaller ponds A2 and A6 were hypoxic, and ponds A1, A3 and A5 – anoxic. The only pond with $O_2$ levels $> 30$ % saturation in September was A4

(Fig. S6). Dissolved oxygen concentrations below ice were not measured, but available literature data clearly indicate that ponds experience anoxic conditions (Matveev et al., 2019).

Dissolved $CO_2$ and $CH_4$ concentrations generally exceeded atmospheric equilibrium irrespective of season and pond characteristics. Similar to DOC, a strongly negative correlation was observed between pond size and concentrations of both $CO_2$ ($p < 0.05$) and $CH_4$ ($p < 0.05$) in September 2024; dissolved $CO_2$ concentrations were significantly lower in large ponds

(Isk and A8 ponds), averaging 30 µM (Fig. S7a). The highest dissolved $CO_2$ concentrations were recorded in the overgrowing ponds A1 and A3, with concentrations of 1740 and 2071 µM, respectively. Although ponds A2, A4, A5, and A6 were to some extent hydrologically connected, they differed in dissolved concentrations of both $CO_2$ and $CH_4$. $CO_2$ concentrations in September ranged from 422 µM in pond A4 to 1010 µM in pond A5 (Fig. S7a). Dissolved $CO_2$ concentrations increased rapidly with the start of the ice formation in October, in some ponds increasing by an order of

magnitude, e.g. in the Isk pond. Winter $CO_2$ concentrations ranged from 2424 µM in pond A3 to 4941 µM in pond A4 (Fig. S7a).

In September, the dissolved $CH_4$ concentrations ranged from 0.4 to 170 µM, with the lowest values found in the stable ponds Isk, A7, and in expanding pond A4 (Fig. 2). The highest concentrations in September, up to 170 µM, were measured in the anoxic ponds A1, A3, and pond A5. Notably, the recently formed pond A6, which is connected to A5, had a mean $CH_4$

concentration that was at least one order of magnitude lower than A5. As with $CO_2$, sampling in early October revealed that all ponds experienced a rapid increase in $CH_4$ concentrations with the start of ice formation with the maximum concentration measured in pond A5 at 701 µM (Fig. 2). In pond A4, dissolved $CH_4$ increased by two orders of magnitude, reaching 144 µM. The increase at Isk pond was smaller than at Áidejávri ponds, with a maximum increase from 1.7 µM to 82 µM (Fig. 2). Dissolved $CH_4$ concentrations in the remaining water column in March ranged from 660 to 1487 µM (Fig. 2). Interestingly,

the overgrowing ponds A1 and A3, which exhibited high concentrations in September and October, showed relatively smaller $CH_4$ accumulation over the winter. In contrast, the highest concentrations were measured in the ponds that were hypoxic in September, with the maximum mean value in pond A2 (1398 µM). The non-thermokarst pond aligned with the others, exhibiting a value of 1302 µM (Fig. 2).





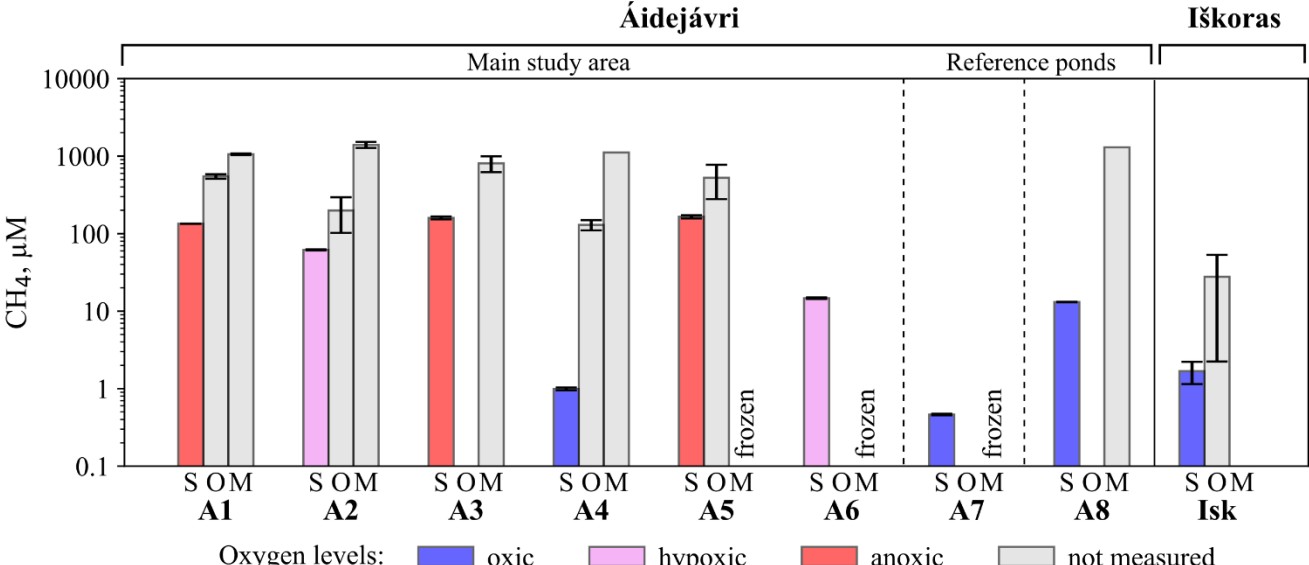

Oxygen levels: ■ oxic ■ hypoxic ■ anoxic ■ not measured

**Figure 2. Dissolved water column CH₄ concentrations in March (M), September (S), and October (O), 2024. Error bars represent the standard deviations of multiple samples collected from different depths in March and October, as well as replicate samples taken in September from the same depth.**

In all ponds, the $CO_2$:$CH_4$ ratio was higher than 1, with the highest values observed in the oxic ponds during the ice-free season in Áidejávri (Fig. S7b). Ratios were also high at Isk pond, both in the ice-free season and under the first ice formed. In most of the ponds, the ratio decreased from September to March, indicating a shift from oxic to anoxic metabolism.

### 4.2. pH, DOC, and CH₄ storage in ice

pH values in the ice samples ranged from 4.4 to 8.2, with an average of 5.9 (Table S1). A pH higher than 7 was only observed in the superimposed ice of non-thermokarst pond A8. DOC concentrations ranged from 1.9 mg L⁻¹ to 160 mg L⁻¹ (Table S1). However, for most ice samples without visible organic material, DOC concentrations were confined to a range of 1.9 to 7.7 mg L⁻¹. Superimposed ice formed a distinct group, with DOC concentrations ranging from 7.4 mg L⁻¹ in the non-thermokarst pond A8 to 42 mg L⁻¹ in pond A6. Values exceeding 60 mg L⁻¹ were only measured in the bottom ice layers containing peat or plants, or within the frozen peat layers (Table S1).

All ice core layers were classified (Sect. 3.2) as "Superimposed ice", "Clear ice", "Methane ebullition bubbles", "Spherical and nut-shaped bubbles", "Elongated bubbles", "Mixed bubbles", and "Frozen peat" (Fig. 3).



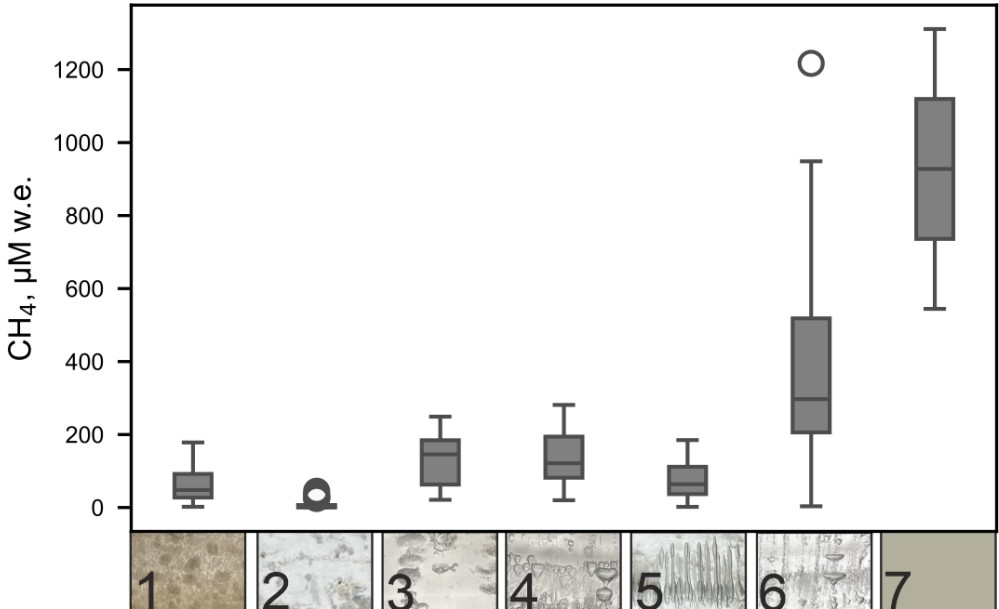

**Figure 3. Box plots illustrating methane ($CH_4$) concentrations in distinct ice types (ice types from Boereboom et al., 2012 with adjustments): 1 – Superimposed ice, 2 – Clear ice, 3 – Methane ebullition bubbles, 4 – Spherical and nut–shaped bubbles, 5 – Elongated bubbles, 6 – Mixed bubbles, 7 – Frozen peat. Error bars are standard deviations across multiple samples within the same ice type category.**

Superimposed ice (1) was found on most ponds, forming an upper, up to 0.1 m thick layer (Fig. 4). In ponds A3 and A6, superimposed ice was up to 0.3 m thick and contained bubbles which may indicate that the ice was indeed formed from pond water (and the contained $CH_4$ thus contributes to the winter $CH_4$ bottom flux, Sect. 3.2). In addition to our visual inspection, a higher DOC content than in ice layers below was another indicator for the presence of superimposed ice derived from pond water (Table S1). The Isk pond had the lowest $CH_4$ content in superimposed ice, averaging 2.9 µM w.e. In Áidejávri, the $CH_4$ content in the superimposed ice ranged from 21 µM w.e. in A3 to 178 µM w.e. in A1.



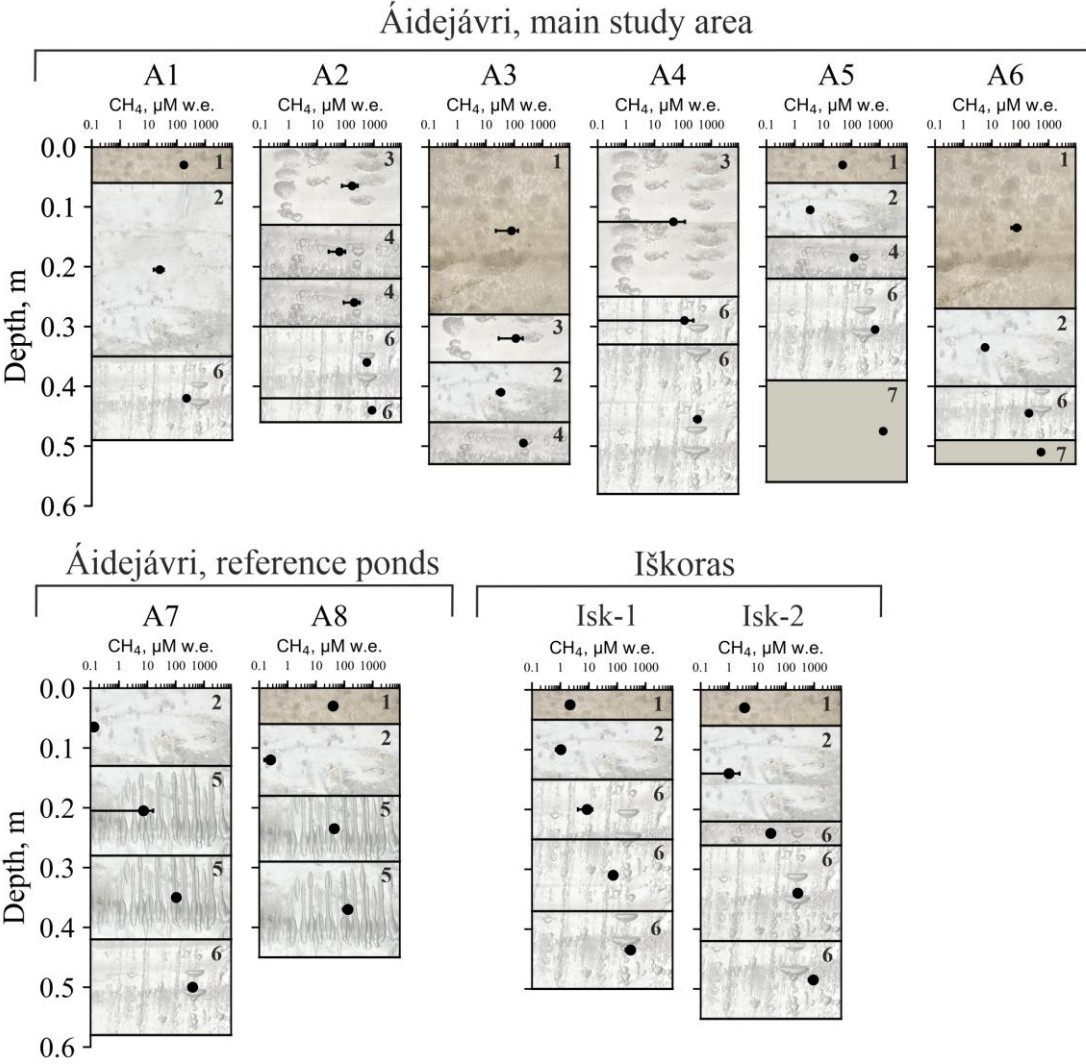

**Figure 4. Ice stratigraphy and methane concentration profiles across various ice and frozen peat layers in ponds from Áidejávri (A1–A8) and Iškoras (Isk–1 and Isk–2). Error bars are standard deviations across multiple samples. The numbers and the background picture of the ice layers correspond to the ice types indicated in the legend in Fig. 3.**

Below the superimposed ice, the $CH_4$ content in the ice increased with depth. Clear ice (2) layers represented the first winter ice formed and were characterized by small, irregularly distributed bubbles with diameters < 2 mm, typically found as the initial ice layer or beneath the superimposed ice. The $CH_4$ content in clear ice varied from 0.1 to 45 μM w.e., with the highest value in the anoxic pond A3.

Methane ebullition bubbles (3), relatively flat and up to 2 cm in diameter, were identified in the surface layers (0 − 0.25 m)
of ice cores from ponds A2 and A4, as well as in the first ice layer formed under the superimposed ice in pond A3 (depth from 0.3 to 0.4 m). The $CH_4$ content of ice layers with methane ebullition bubbles varied between 21 and 249 μM w.e. Distinct layers of spherical and nut–shaped bubbles (4) were found in only four ice cores: A2, A3, and as individual thin



layers in A5 and Isk-2 (Fig. 3). $CH_4$ content of these bubbles ranged from 20 to 281 µM w.e. within the same ice column of pond A2, but at different depths. In pond A3, this type was found the middle ice layer, with an average $CH_4$ content of 212

µM w.e. Elongated bubbles (**5**) were present only in two large ponds in Áidejávri: the old thermokarst pond A7 and the non–thermokarst pond A8. The $CH_4$ content in the ice of these ponds increased with depth, ranging from 1.9 to 122 µM w.e. in A8 and from 43 to 185 µM w.e. in A7.

When spherical, nut–shaped, elongated, and methane ebullition bubbles were found together in an ice layer, it was attributed to the mixed bubbles group (**6**), typically located in the lower parts of the ice core. This group was identified in all ice cores

except for A3 and A8 and exhibited the highest $CH_4$ content among the ice types (Fig. 3). The minimum (3.4 µM w.e.) and maximum (1217 µM w.e.) $CH_4$ content for the mixed bubbles group, spanning all sites, were recorded in the parts of Isk pond (Isk-1 and Isk-2, respectively). The concentrations in the deepest layer of the old (Isk-1) and young (Isk-2) parts of the Iškoras pond were significantly different, with an average $CH_4$ content of 933 µM w.e. in ice core Isk-2 (young) and 299 µM w.e. in ice core Isk-1 (old). In Áidejávri, the recently formed pond A6 and the deep sedge–growing pond A1 exhibited the

lowest average $CH_4$ content in the mixed bubble layers around 200 µM w.e., respectively. In contrast, pond A5, despite its connection to A6 and similar depth, had a more than three times higher methane content. The $CH_4$ content of the mixed bubble layers increased with depth from 567 to 856 µM w.e. in A2, and from 114 to 321 µM w.e. in A4. The shallow ponds A5 and A6 (each with a depth of less than 0.4 m) were completely frozen to the bottom, including the uppermost peat bottom layer. Within the frozen peat (7), the measured $CH_4$ content were 1311 µM w.e. for pond A5 and 544 µM w.e. for pond A6.

**4.3 $CH_4$ winter storage**

The largest contribution of ice storage to total winter $CH_4$ storage, up to $5010 \pm 473$ mg $CH_4$-C m⁻², was found in ponds that were fully or nearly frozen to the bottom, as observed in the Áidejávri pond A5 and the young part of the Isk pond (Isk-2). In contrast, other shallow ponds that were completely frozen to the bottom – such as A6, A7, and Isk-1 – exhibited lower values, from $443 \pm 221$ to $854 \pm 102$ mg $CH_4$-C m⁻², highlighting the significance of thermokarst pond formation age (Sect.

5.4) as a contributing factor. Little $CH_4$ storage in the ice was observed in the deeper ponds A1, A3, and A8 (Table S2), while the expanding deep thermokarst ponds A2 and A4 showed significantly more $CH_4$ storage in ice with values of $1580 \pm 259$ mg $CH_4$-C m⁻² and $1120 \pm 147$ mg $CH_4$-C m⁻², respectively, standing out from the patterns described for the other ponds. The uncertainty of the ice storage estimation did not exceed 30 % in most of the ponds, being highest with 50 % in the ice core A6 with a significant contribution of superimposed ice in the ice column (Sect. 3.5).

Below-ice $CH_4$ storage was highest in the deep ponds A1 and A2, reaching $16626 \pm 1874$ mg $CH_4$-C m⁻² in the pond A2. In the Áidejávri ponds A3, A4, and A8, values ranged from $3347 \pm 1349$ to $5524 \pm 1008$ mg $CH_4$-C m⁻² (Table S2). The uncertainty in these estimations was mostly related to depth error and was less than 40 % for the Áidejávri ponds. The below-ice storage calculated for the Isk pond was 636 mg $CH_4$-C m⁻² for Isk-2 part and 1273 mg $CH_4$-C m⁻² for Isk-1 part, with a 100 % uncertainty due to the ambiguity of whether a residual water layer remained below the ice (Sect. 3.5).



To estimate the $CH_4$ storage prior to freezing, we use both measurements from September 2024 and from the first winter ice layer (Sect. 3.5). In the latter, the $CH_4$ contents (Sect 4.2) were up to 5 times lower than the dissolved $CH_4$ concentrations measured in September 2024 for most of the ponds (Fig. S8), with the exceptions of A5 and A8 where it was 50 times lower. The largest $CH_4$ dissolved storage before ice growth was measured in the deepest ponds A1, A2, and A3, with values ranging from $1174 \pm 86$ to $1364 \pm 881$ mg $CH_4$-C m⁻² (Table S2). At the same time, the shallow pond A5 exhibited a storage of $419 \pm 415$ mg $CH_4$-C m⁻², while the remaining ponds showed values not exceeding 70 mg $CH_4$-C m⁻² (Table S2). The initial storage of $CH_4$ prior to freezing introduces an uncertainty of 5 % to 100 %. However, considering the comparatively low pre-freezing storage relative to the $CH_4$ storage in and below the ice, this error does not contribute strongly to the final storage uncertainty.

**4.4 Winter $CH_4$ bottom flux**

The winter $CH_4$ bottom fluxes (Sect. 3.5) for thermokarst ponds in Áidejávri varied significantly, ranging from $2.8 \pm 1.4$ mg $CH_4$-C m⁻² d⁻¹ in the recently formed shallow pond A6 to $107 \pm 14$ mg $CH_4$-C m⁻² d⁻¹ in the pond A2 formed 40 years ago (Fig. 5, Table S2). In comparison, the fluxes at the Isk pond were lower than those observed in Áidejávri with similar formation age (Fig. 5). Even assuming that the Isk pond was not frozen to the bottom and adding a potential below-ice $CH_4$ storage (Sect. 3.5), the fluxes were estimated to be $11 \pm 8$ mg $CH_4$-C m⁻² d⁻¹ in the old part of the pond (Isk-1) and $15 \pm 5$ mg $CH_4$-C m⁻² d⁻¹ in the section of the pond formed between 20 and 68 years ago (Isk-2).

A significantly higher $CH_4$ flux was estimated for the non-thermokarst pond A8 with $36 \pm 10$ mg $CH_4$-C m⁻² d⁻¹ in comparison to the thermokarst ponds of the same size. The uncertainty in the calculated fluxes was less than 31 % for all ponds, except for pond A6, where the presence of superimposed ice resulted in a 50 % uncertainty in the estimated flux, and the oldest part of Isk pond Isk-1, where uncertainty in below-ice $CH_4$ water storage led to a higher value of 72 % (Sect. 3.5).



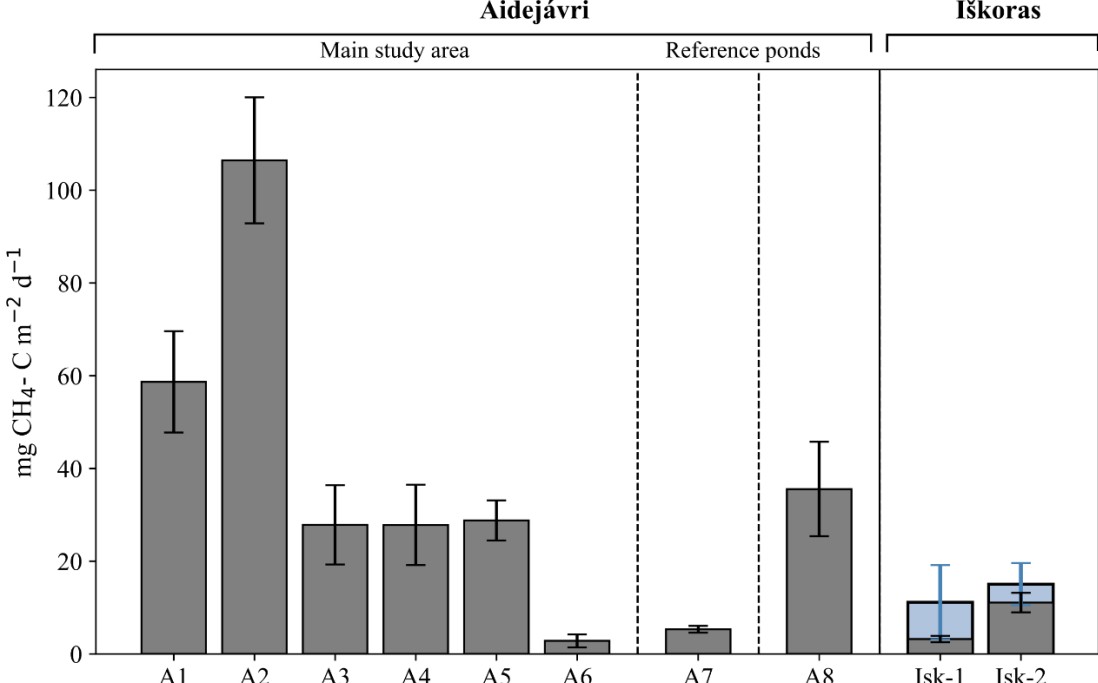

365

**Figure 5. CH₄ winter bottom flux in the studied ponds. Error bars represent the uncertainties calculated using Gaussian error propagation (Sect. 3.5). Blue error bars indicate the values estimated based on the assumption of below-ice dissolved methane storage in the Isk pond (Sect. 3.5) The vertical lines separate the main study area in Áidejávri (A1–A6) from the reference ponds A7 and A8, and the Isk pond.**

370 There was a strongly positive correlation ($p < 0.05$) between pond depth and winter bottom CH₄ flux, resulting in the highest fluxes estimated for lakes A1 and A2. However, this pattern does not explain the similar fluxes observed in ponds A3, A4, and A5, which have different depths. For ponds A1-A6, which originated form the same section of the peat plateau (Sect. 2), we compare the relationship between the age of thermokarst pond formation and winter CH₄ bottom fluxes (Fig. 6). We observed an increase in winter CH₄ bottom flux with age, from about 3 mg CH₄-C m⁻² d⁻¹ in the recently formed pond A6 to

375 around 30 mg CH₄-C m⁻² d⁻¹ in ponds A3, A4, and A5, which were formed between 3 and 20 years ago. The highest flux was recorded in the approximately 40-year-old pond A2. The oldest pond A1 had a smaller CH₄ bottom flux than A2, but still higher than all younger ponds. While the statistical evaluation of this age relationship is challenging due to the limited number of ponds, the relationship between pond age and bottom flux is close to being statistically significant ($p = 0.07$).





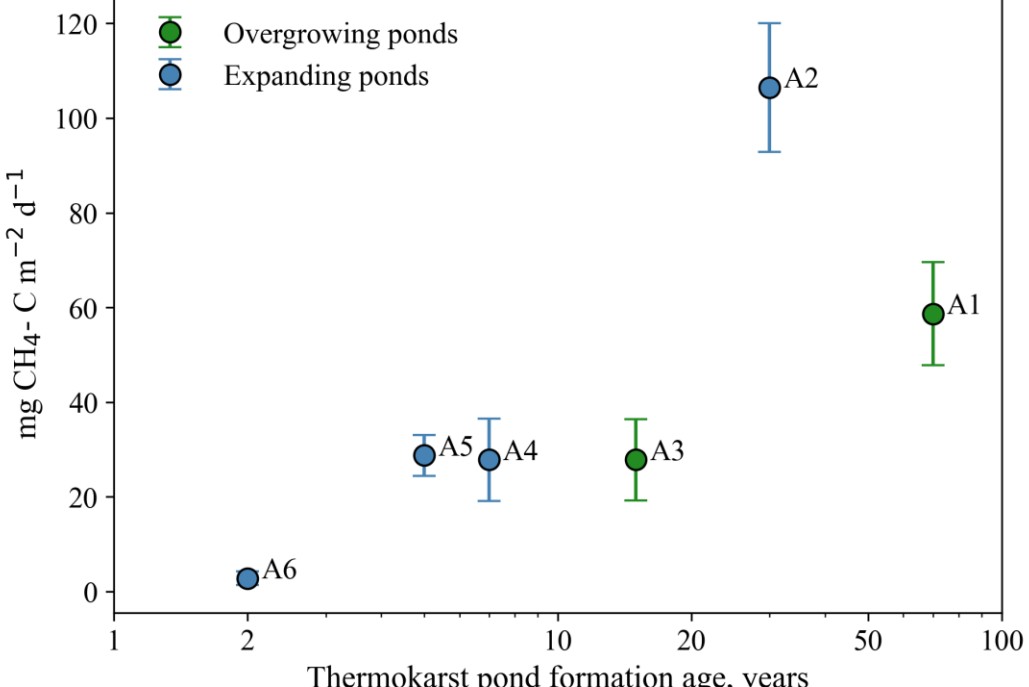

**Figure 6. Winter CH₄ bottom flux vs. thermokarst pond formation age in ponds of the main study area in Áidejávri (A1-A6).**

## 5 Discussion

### 5.1 Methodological limitations

We estimated winter bottom CH₄ flux for the 2023–2024 winter season for nine ponds located within two permafrost peatlands in Northern Scandinavia. We based our estimates on methane storage in the ice (ice bubble storage) and the unfrozen water column below, corrected for the storage prior to freezing. Conceptually, we used the pond ice and the remaining water as a natural chamber to quantify winter methane bottom flux which captures both ebullition and diffusive fluxes of CH₄. Our approach entails the following uncertainties and limitations:

– A significant uncertainty arises from measuring the depth of the water column in the shallow ponds, especially under the winter ice. The loose peat material and presence of plants at the bottom renders accurate measurements of pond depth challenging (Sect. 3.5). We accounted for this by including an estimated uncertainty of 10 to 100 % of the water depth, with 100 % for thin water layers of only 0.05-0.1 m thickness under the ice. Moreover, sampling was conducted at the central part of the ponds, but lateral variations in pond depth could lead to biases in the CH₄ storage accumulated during the winter, particularly in the larger ponds. The presence of plants on the bottom and difficulties in sampling dissolved CH₄ from the residual water layer (Sect 3.5) resulted in high flux uncertainties of about 70 % for the Isk pond locations Isk-1 and Isk-2.





- The upper 0.05–0.1 m of the ice column of most of the frozen ponds was composed of superimposed ice (Sect. 4.2), with nearly half of the column composed of this ice for two of the cores. This type of ice forms primarily by freezing of melt- or rainwater within the snowpack on the ice surface. However, it can also be formed when unfrozen water spreads across the ice surface following crack formation and subsequently refreezes, resulting in high DOC contents in the ice (Manasypov et al., 2015). In the first case, any stored $CH_4$ would originate from the ambient air, thus not contributing to the bottom $CH_4$ flux. In the latter case, the $CH_4$ trapped in the ice originates at least partly from the water body below. Since we cannot distinguish between these cases, we excluded the superimposed ice layer from our $CH_4$ storage calculations to avoid overestimating $CH_4$ flux. In any case, $CH_4$ contents in the superimposed ice were relatively low compared to those in the deeper ice layers and the dissolved $CH_4$ storage in the water column. For almost all cores, the contribution of superimposed ice to the total $CH_4$ storage was less than 5 % and thus was negligible in the final uncertainty of winter bottom fluxes. However, for pond A6, superimposed ice comprised half of the ice column, resulting in a contribution of 50 % which was included in the final uncertainty. Therefore, the winter $CH_4$ bottom flux of A6 features a high relative uncertainty, but it is still clear that the absolute flux values are significantly below those of the other ponds A1-A5 (Fig. 6, Sect. 4.4).

- Another source of uncertainty is related to the lack of measured concentrations of dissolved $CH_4$ prior to freezing in 2023. To estimate initial $CH_4$ storage, we used the average concentration from the first winter ice layer combined with available September data from 2024. In some cases, this approach led to an uncertainty of up to 100 % for the $CH_4$ storage prior to freezing. However, this uncertainty has a limited effect on the final $CH_4$ flux estimates, as the initial $CH_4$ storage was relatively small compared to the total $CH_4$ accumulation during winter. For example, $CH_4$ storage after winter accumulation was 4 to 200 times higher than the initial $CH_4$ storage. Notably, in ponds that were anoxic before freezing, the differences between initial and accumulated $CH_4$ storages (Fig. S8) were less pronounced (4 to 14 times).

Despite the challenges in estimating the individual terms of the $CH_4$ storage, our approach captures both ebullition and diffusive fluxes during the winter, thus making it well-suited to estimate wintertime $CH_4$ storage in high latitudes. In this study, we have estimated the uncertainties of the individual terms of the $CH_4$ winter storage, including ice storage, below-ice water storage and the $CH_4$ storage prior to freezing, and used Gaussian error propagation to determine an overall uncertainty for the winter $CH_4$ bottom flux. Our ice records suggested that $CH_4$ production in shallow ponds continued until the pond was frozen to the bottom in mid-March (Fig. S5). This highlights the importance of assessing both the ice storage and the water storage under the ice. In ponds shallower than 0.6 m, the ice storage is the dominant term for methane winter storage, while the ice storage accounted for less than 25 % of the estimated $CH_4$ storage in deeper ponds. The absolute magnitude of these individual storage terms strongly influences the final uncertainty, with the uncertainty in shallow ponds largely related to ice storage, while the below-ice water storage dominates the uncertainty for deep ponds.





## 5.2 Annual CH$_4$ budget at the Iškoras site

In Finnmark, shallow ponds are ice-covered for at least seven months in a year. Anoxic conditions are likely to establish
shortly after the formation of the first winter ice. The lack of oxygen and the reduced activity of methanotrophs at low
temperatures lead to accumulation of CH$_4$ in and under the ice, which is released to the atmosphere during ice-off in late
spring (Denfeld et al., 2018). This highlights the importance of accurately estimating wintertime CH$_4$ production and storage
to understand and predict the CH$_4$ budget of northern water bodies.

Assuming a constant production rate and 228 days of ice cover prior to ice melt on May 22, 2024 (Fig. S4), we estimate the
wintertime CH$_4$ storage in the ponds to be 0.6–24 g CH$_4$-C m$^{-2}$ for Áidejávri and 3 g CH$_4$-C m$^{-2}$ for Iškoras. These estimated
storage values in our study are generally higher than those reported for small peatland lakes (Kuhn et al., 2021) and
thermokarst lakes in Alaska (Sepulveda-Jauregui et al., 2015). However, they align with CH$_4$ storage values accumulated
until ice melt for thermokarst lakes at a Canadian palsa site, which were around 5 g CH$_4$-C m$^{-2}$ (Matveev et al., 2019).

At Iškoras, methane flux is measured by an eddy covariance tower, and a footprint analysis combined with machine learning
has delivered average CH$_4$ fluxes for the thermokarst ponds within the peat plateau complex (Pirk et al., 2024). As the Isk
pond investigated in this study comprises a large fraction of the "thermokarst pond" class within the eddy covariance
footprint, it is possible to establish a rough annual CH$_4$ balance by combining the eddy covariance measurements during the
ice-free season with winter-time CH$_4$ bottom fluxes. For the ice-free season, the mean CH$_4$ flux for ponds measured by the
eddy covariance system is 38 mg CH$_4$-C m$^{-2}$ d$^{-1}$ (Pirk et al., 2024; Table S3), while our mean daily winter-time CH$_4$ bottom
flux for the Isk pond is about three times lower (13 mg CH$_4$-C m$^{-2}$ d$^{-1}$). This is consistent with findings from Alaskan lakes,
where summer CH$_4$ production rates increase by a factor of two to three due to warmer bottom sediment temperatures
(Matheus Carnevali et al., 2015). The apparent Q$_{10}$ value for the estimated CH$_4$ bottom flux at Iškoras is 2.7, based on a
mean bottom water temperature of 1°C under ice cover and an average temperature of 11°C during the ice-free period (Fig.
S5, Table S3). This value aligns well with previously reported Q$_{10}$ values for northern lakes (Kuhn et al., 2021).

Assuming that Áidejávri ponds follow the same Q$_{10}$ relationship as inferred at Iškoras, the corresponding ice-free methane
flux for Áidejávri would range from 8 to 300 mg CH$_4$-C m$^{-2}$ d$^{-1}$, with an average value of 103 mg CH$_4$-C m$^{-2}$ d$^{-1}$. This
assumption allows us to estimate the annual CH$_4$ budget for each pond system, which ranges from 2 to 66 g CH$_4$-C m$^{-2}$
year$^{-1}$, with averages (over all ponds) of 8.3 and 23 g CH$_4$-C m$^{-2}$ year$^{-1}$ at Iškoras and Áidejávri, respectively.

Our back-of-the-envelope calculations suggest that winter methane accumulation at Iškoras constitute up to 40 % of the
yearly flux in the ponds. Therefore, the resulting CH$_4$ storage likely results in elevated emissions during the ice-melt period.
Multi-year data from the Iškoras eddy covariance tower showed only small methane flux peaks during the ice-melt period. A
direct comparison of these pond flux estimates is difficult, as the eddy covariance flux footprint only comprises about 7 %
pond surfaces on average, and the wind direction only seldomly moved the footprint to pond surfaces during the ice-melt
period. Hence, a dedicated experimental design would be required to reliably capture the CH$_4$ ice-off signal in such complex
landscapes. In some northern lakes, ice-melt emissions have been reported to contribute up to 60 % of the annual CH$_4$ budget



(Jansen et al., 2019; Denfeld et al., 2018). By contrast, the CH$_4$ ice-free emissions from thermokarst ponds in the sporadic permafrost zone of the West Siberian Lowland (WSL) did not differ significantly from ice-off emissions (Serikova et al., 2019). Interestingly, in areas of continuous permafrost, diffusive CH$_4$ flux during the ice-off and ice-free periods showed pronounced variability (Serikova et al., 2019). This highlights the need for additional observational data for small peatland
ponds to better constrain the magnitude of CH$_4$ fluxes during ice melt.

### 5.3 Key factors controlling the CH$_4$ budget

The wintertime CH$_4$ flux is smaller than the summertime flux, partly caused by temperature differences in the sediment between winter and summer (e.g. Fig. S5). Furthermore, the winter microbiome in thermokarst lake water was found to differ significantly from the summer microbiome in both microbial composition and metabolic functions, with taxa
supporting multiple reductive pathways dominating the winter microbiome and enhancing the degradation of permafrost-derived organic matter (Vigneron et al., 2019). This taxonomic shift in the C cycling microbiome goes together with a general shift in oxygen availability; while pO$_2$ may play an important role for CH$_4$ formation and oxidation in summer, thermokarst ponds are generally anoxic in winter. Many small ponds in Áidejávri area exhibited anoxic conditions as early as September (Sect. 4.1, Fig. S6), suggesting that CH$_4$ oxidation is marginal during the ice-free period. By contrast, ponds
larger than 500 m² were O$_2$ saturated during this period (Sect. 4.1, Fig. S6). In small ponds, CH$_4$ oxidation likely does not play a significant role for the overall CH$_4$ budget, while in larger ponds CH$_4$ oxidation might occur at the beginning of the freezing season. However, the effect on the overall CH$_4$ balance is likely small, as oxygen would be rapidly depleted after an ice cover has formed.

In our study, the highest CH$_4$ flux was estimated for thermokarst ponds smaller than 500 m². This is in agreement with
previous studies that found that pond size is a critical factor for CH$_4$ fluxes during the ice-free period (Bastviken et al., 2004; Shirokova et al., 2013; Zabelina et al., 2021, Manasypov et al., 2024). Among the key factors driving CH$_4$ production in our dataset, CH$_4$ winter bottom flux showed a strongly positive correlation with the pond depth, with deeper lakes consistently exhibiting a higher CH$_4$ bottom flux.

The CH$_4$ bottom fluxes observed in Áidejávri ponds were generally higher than those in the Iškoras pond. This aligns with
the results of the incubation experiments under controlled temperature and anoxic conditions of permafrost samples from these sites (Kjær et al., 2024). The CH$_4$ production potential of the Iškoras permafrost samples was lower than for the samples from Áidejávri, which might be related to differences in peat formation history, resulting in different geochemistry of the peat (e.g. a high iron content in Áidejávri) or other local factors (Kjær et al., 2024).

The CH$_4$ winter bottom fluxes in the studied ponds in Finnmark ranged from 3 to 107 mg CH$_4$-C m$^{-2}$ d$^{-1}$. While data on
wintertime CH$_4$ bottom flux from thermokarst ponds remain limited, our findings are within the range of winter fluxes reported for other northern regions. In polygonal ponds in the Lena River Delta (Eastern Siberia), winter CH$_4$ fluxes ranged from 0.01 mg CH$_4$-C m$^{-2}$ d$^{-1}$ in ponds at the initial development stage to 104 mg CH$_4$-C m$^{-2}$ d$^{-1}$ in ponds exhibiting signs of thermal erosion (Langer et al., 2015). The mean winter daily flux estimates for glacial lakes in Northern Sweden ranged from



0.001 to 9.5 mg $CH_4$-C $m^{-2}$ $d^{-1}$ which is at the lower limit of the fluxes measured in Finnmark. This comparison renders

thermokarst ponds, particularly those formed in organic-rich regions such as yedoma and permafrost peatlands, to be significant sources of $CH_4$ compared to other lake types in northern ecosystems (Wik et al., 2016; Kuhn et al., 2021). The relatively higher $CH_4$ fluxes in thermokarst systems are related to the input of permafrost-derived organic matter, which accelerates $CH_4$ production rates. In our study, we observed a strongly negative correlation between pond size, dissolved $CH_4$, and $CO_2$ concentrations and DOC at the end of the ice-free season in September (Sect. 4.1). Higher DOC

concentrations enhance oxygen consumption, leading to oxygen depletion or complete anoxia, which favors methanogenesis and $CH_4$ release with minimal oxidation during the summer. Together with the predominance of ebullition as a major emission pathway in small thermokarst peatland ponds (Kuhn et al., 2018), this suggests that most of the $CH_4$ produced during the summer is released to the atmosphere. Factors controlling the production of $CH_4$ are likely similar in both winter and summer, with seasonal differences largely driven by variations in sediment temperature.

**5.4 Winter $CH_4$ bottom flux vs. thermokarst pond formation age**

The $CH_4$ production in thermokarst water bodies is influenced by a number of factors, including permafrost type (organic-rich vs. non-organic-rich), pond depth and size, as well as vegetation (Burke et al., 2019). Our study design makes it possible to explore the role of the age of thermokarst pond for $CH_4$ production. For this purpose, we focused on the ponds in the main study area of Áidejávri (A1-A6, Sect. 2) which shared the same peat properties, vegetation, and hydrological regimes, while

exhibiting similar DOC concentrations (ranging from 90 to 105 mg $L^{-1}$). Our results suggest that there may be a functional relationship between the winter $CH_4$ bottom flux and the thermokarst pond formation age. Thermokarst lakes and ponds undergo several development stages, from initial thawing and expansion to succession into wetland ecosystems. Methane production is generally strongest during the first decades of thermokarst pond evolution when permafrost thawing is most active (Walter et al., 2006; Desyatkin et al., 2009; Shirokova et al., 2013, Vonk et al., 2015). However, in our study, the

youngest pond, A6 (2 years old), exhibited the lowest winter $CH_4$ bottom flux among all the studied lakes. Meanwhile, the pond A5, which is only 2 years older, showed the flux 10 times higher. This delayed onset of $CH_4$ production could possibly be explained by the lag phase required for methanogenic archaea to adapt and recover after prolonged dormancy in frozen permafrost to new environmental conditions (Rivkina et al., 2007; Knoblauch et al., 2018). This lag is also associated with the shift from previously oxic conditions in the active layer to newly established anoxic conditions conducive to

methanogenesis. Such lag phases in $CH_4$ production are well-documented from incubation studies conducted with Siberian permafrost (Rivkina et al., 2007, Knoblauch et al., 2018) and from the same peatlands in Finnmark studied here (Kjær et al., 2024). Similar to our results, a low $CH_4$ production during the initial stages of thermokarst pond formation has also been estimated for the polygonal tundra ponds of the Lena River delta (Langer et al., 2015). An alternative explanation for the low $CH_4$ winter bottom flux in the only two years old pond A6 could be that more $CH_4$ is indeed produced but is trapped below

the still intact root zone of the freshly submerged peat plateau surface, thus preventing its release to the water column. During summer, we observed that a large amount of gas bubbles could be released by disturbing this uppermost root layer



with a probe which suggests that gas is indeed accumulated. While the exact dynamics of this process remain unclear, it is at least possible that not all the produced $CH_4$ is directly released to the water column and eventually the atmosphere in such young thermokarst water bodies.

Pond A2 shows the highest winter $CH_4$ bottom flux among all studied ponds, while the younger pond A4, which shares similar physical characteristics (i.e., pond depth, location, and evidence of ebullition bubbles in ice column), exhibits fluxes almost four times lower. This difference may indicate that thermokarst pond formation age is a major factor influencing $CH_4$ production, when other environmental conditions are comparable.

Methane production and emissions from newly formed thermokarst lakes or along thermokarst margins in organic-rich
permafrost areas are generally higher than those from older ones (Vonk et al., 2015; Walter Anthony et al., 2016; Heslop et al., 2020). In the Iškoras pond, the estimated winter bottom $CH_4$ flux from the younger part of the pond was at least 30 % higher than from the central part older than 68 years. In the main study area of Áidejávri (A1-A6), our results show that $CH_4$ production can remain high in ponds older than 60 years. For example, the winter bottom $CH_4$ flux was about 60 mg $CH_4$-C $m^{-2}$ $d^{-1}$ for the old pond A1, which is more than double the values found for the relatively young thermokarst ponds (A3-
A6).

While the limited number of investigated ponds does not allow a robust statistical evaluation, ponds in the later stages of development during the transition to a permafrost-free mire (A1, A3, A7, Isk) displayed significant differences in $CH_4$ wintertime production which may be related to vegetation type. Ponds undergoing a transition through sedge regrowth exhibited higher $CH_4$ winter bottom fluxes than the ones experiencing Sphagnum colonization. Sedges release highly labile
organic acids, such as root exudates, including sugars and organic acids, which are rapidly converted into $CH_4$ by methanogens (Ström et al., 2012; Dorodnikov et al., 2011). Furthermore, vascular plants can reduce oxidation rates and deplete oxygen levels, particularly toward the end of the growing season (Turner et al., 2020). In contrast, the old, stable lakes at both Iškoras and Áidejávri (Isk and A7) in which Sphagnum mosses were strongly growing at the edges displayed low winter methane bottom flux (< 15 mg $CH_4$-C $m^{-2}$ $d^{-1}$). The presence of the Sphagnum mosses may directly reduce $CH_4$
emissions due to methane oxidation processes facilitated by symbiotic relationships between Sphagnum mosses and methane-consuming endophytic bacteria (Raghoebarsing et al., 2005), as well as the presence of aerated surface peat layers (Parmentier, Huissteden, et al., 2011; Magnusson et al., 2020).

**Conclusions**

This study assesses wintertime methane ($CH_4$) flux in nine shallow ponds within two permafrost peatlands in Northern
Norway (Iškoras and Áidejávri) during the 2023–2024 winter season. These ponds form a chronosequences of thermokarst formation spanning more than 70 years. Using pond ice and the below-ice water column as natural chambers, we estimate winter $CH_4$ bottom flux and storage. Our key findings are:



- At the study sites, ponds remained ice-covered for more than seven months per year. Limited oxygen availability and reduced methanotrophic activity led to $CH_4$ accumulation. The average wintertime $CH_4$ bottom flux ranged from 3 to 107 mg $CH_4$-C $m^{-2}$ $d^{-1}$ across the studied ponds in the two permafrost peatlands.

- The wintertime $CH_4$ storage ranges from 0.6 to 24 g $CH_4$-C $m^{-2}$ which at the Iškoras site constitutes up to 40 % of the annual $CH_4$ budget.

- Ponds dominated by sedge regrowth have larger $CH_4$ bottom fluxes, while older ponds experiencing colonization by Sphagnum mosses feature smaller $CH_4$ bottom fluxes.

- Pond age appears to be a significant factor influencing the wintertime $CH_4$ bottom flux. A young pond that formed two years prior showed the smallest flux, but fluxes increased considerably with pond age. Our results suggest that the wintertime $CH_4$ bottom flux can remain at high values (up to 60 mg $CH_4$-C $m^{-2}$ $d^{-1}$) in thermokarst ponds older than 70 years.

The current atmospheric warming trends may lead to an acceleration of thermokarst processes, resulting in an increase of small shallow thaw ponds in organic-rich permafrost regions. While this could increase $CH_4$ emissions from these permafrost landscapes, studies on the seasonality and especially the wintertime production of $CH_4$ remain sparse. This study highlights the need to constrain the wintertime $CH_4$ production from thermokarst ponds to accurately estimate the present-day and project the future $CH_4$ budgets at high latitudes.

*Data availability.* Data supporting this study will be made available in the Zenodo permanent repository upon acceptance (https://zenodo.org).

*Author contributions.* AP, SW and PD conceptualized the research. AP, SW, MRI, CW and SS conducted the field sampling. AP conducted the laboratory analyses, with PD providing expertise and help. AP performed the data analysis, with input from PD, NP and SW. NP contributed with data from eddy covariance measurements in Iškoras. AP prepared the manuscript, and all co-authors revised and edited the final version.

*Competing interests.* The contact author has declared that none of the authors has any competing interests.

*Acknowledgements.* We would like to thank our colleagues at the Faculty of Environmental Sciences and Natural Resource Management, NMBU, especially Thomas Rohrlack, Sigrid Trier Kjær, Trygve Fredriksen and Mona Mirgeloybayat. ChatGPT (version GPT-4) was used for English grammar and spelling correction during the preparation of the manuscript.

*Financial support.* This study was supported by BIOGOV (project no. 323945, Research Council of Norway) and PEAT-THAW (within the sustainability initiative of the Faculty of Mathematics and Natural Sciences, University of Oslo, Norway).





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
