# Peer review of "Wintertime Production and Storage of Methane in Thermokarst Ponds of Subarctic Norway"

_EGUsphere, 2025_

## Author Comment (AC2)

**RC1: 'Comment on egusphere-2025-3059', Anonymous Referee #1, 09 Sep 2025**

We thank the reviewer for taking the time to review our paper. The comments helped us to improve the manuscript. We fully restructured the sections on Study area and Methods and shorten the Discussion following the reviewer's recommendations. We also removed inaccurate wording. In general, there are only few studies that quantify wintertime $CH_4$ in thermokarst lakes, and even fewer from thermokarst ponds in permafrost peatlands (e.g., Kuhn et al., 2021). However, such data are very important to understand the annual $CH_4$ budget of such ponds which again is a prerequisite for upscaling to permafrost landscapes and beyond (e.g. Bastviken and Johnson, 2025). Our study provides over-winter $CH_4$ flux measurements from nine ponds in Northern Scandinavia. One of the study areas includes an eddy-covariance tower, which has been used to estimate the contribution of $CH_4$ accumulated in thermokarst ponds during winter to the annual budget. In addition to quantifying $CH_4$ wintertime flux and storage in thermokarst ponds, our study design linked to repeated aerial imagery allows us to estimate relationships between $CH_4$ production and thermokarst pond formation age.

In the following, we respond to the issues raised by the reviewer and indicate where we implemented changes in the revised manuscript. Reviewer comments appear in black, our responses appear in blue, and revised manuscript text appears in *blue italics*.

**Comments**

**Revie of "Wintertime Production and Storage of Methane in Thermokarst Ponds of Subarctic Norway"**

Here, Pismeniuk et al. quantified methane storage and emissions from several thermokarst peatland ponds during the ice covered period. By using chronosequences of thermokarst pond formation, they explained the observed rates and distinguished them according to vegetation types. The topic fist very well for Biogeosciences. However, at this stage, the manuscript contains several inaccuracies that need to be addressed. Please see my main comments below. The primary aim of the study is to quantify methane emissions and storage in ponds over time. I acknowledge the logistical challenges of sampling in remote locations, particularly given the number of ponds included. However, for the study it is necessary statistical strength to talk about ecosystem level replicates. For instance, in the case of pond A8, it is unclear why this very particular case was included. Please provide a stronger justification for its inclusion or consider removing it from the study. Regarding thermokarst formation, your results suggest a promising pattern. However, in several cases there are no replicates. For example, A6 represents a recently formed pond, but no comparative sites are provided, while A3 appears to present a similar issue. Therefore, you need to clearly explain the rationale behind your pond selection.

Our study does not really aim to draw conclusions at the ecosystem level (e.g. for the entire peat plateau complex), but we specifically focus on the ponds as potential $CH_4$ hotspots within the larger-scale ecosystem (e.g. Vonk et al., 2015; Kuhn et al., 2021). Our objectives are to (1) quantify

CH4 winter bottom fluxes in various ponds across two permafrost peatland sites in Finnmark, (2) assess the contribution of wintertime cumulative CH4 flux to the annual CH4 budget at one of the sites, (3) identify the main factors causing differences in CH4 winter bottom fluxes, and (4) explore the relationship between CH4 winter bottom fluxes and the age of thermokarst pond formation. These objectives are now clearly stated in the Introduction.

Our data from the two peat plateaus show that methane bottom fluxes differ significantly between Iškoras and Áidejávri, which supports the conclusion that site-specific factors at least to some degree govern CH4 production across different permafrost peatlands. We consider this an important finding as it complicates the upscaling of CH4 fluxes to the ecosystem scale and beyond (Sect. 5.3, revised manuscript). For this reason, we do not draw any conclusion on fluxes on the ecosystem scale.

In the main study area (ponds A1–A6,) we include all available ponds formed from the same peat plateau segment, covering an area of 150 m × 100 m. The formation from the same peat plateau segment suggests that the submerged peat material has similar characteristics, while also the meteorological conditions and the hydrological regime are similar. This makes it possible evaluate the influence of pond formation age with limited confounding factors. The ponds A6 and A3 are the only available ponds of their respective ages. In the revised manuscript (Sect. 2.2), the rationale for selecting ponds A1–A6 reads: *"Six of the ponds (A1-A6) are located in a relatively small area of 150 m x 100 m in which all available ponds were sampled (referred to as "main study area" in Áidejávri in the following, see also Fig. S1). These ponds were all derived from the same peat plateau complex, suggesting similar environmental conditions and a shared origin of the submerged peat material. However, analysis of air photos (Sect. 3.7) revealed different formation ages of the ponds, spanning a chronosequence from 1 to 70 years, which allows us to compare wintertime CH4 production with thermokarst pond formation age (Fig. S1)."*

Pond A8 is included because it differs in origin and characteristics from the thermokarst ponds. Unlike the thermokarst ponds in Áidejávri, A8 has an elongated shape and a markedly different pH. We interpret A8 as a remnant of a larger post-glacial water body that is partly transitioning to mire through sedge succession. Including pond A8 places the calculated CH4 fluxes for thermokarst ponds in the main study area in a broader context and delivers value for future comparisons.

At the Iškoras peat plateau complex, we sampled two locations in the large central pond. This pond is situated in the footprint of the eddy-covariance tower operating at Iškoras. Pirk et al. (2024) disintegrated the fluxes of the eddy-covariance tower into different landcover types, one of which are "ponds". The studied pond is by far the largest in the footprint, so that the values for the "pond" class can be used as a proxy for cumulative summer fluxes of this pond. This puts us in the position to estimate the contribution of cumulative wintertime flux to the annual CH4 budget (Sect. 5.2, revised manuscript). In general, there are fewer ponds in Iškoras compared to Áidejávri, as the total area of the peat plateau complex is much smaller. This makes it challenging to establish a

similar thermokarst pond chronosequence at Iškoras. During the fieldwork in March 2024, we tried to sample some of the younger ponds, but these proved to be very shallow, and sampling was unsuccessful.

In the revised manuscript, we have rearranged the Study Area section and now include a more detailed description of the study site and the rationale behind the pond selection (Sect. 2.2, Sampling sites, revised manuscript). We also added a section on thermokarst age evaluation in Methods and Supplementary material (Sect. 3.7, Fig. S1-3, revised manuscript).

The protocol for estimating $CH_4$ storage in the ponds is confusing, as it relies on arbitrary or insufficiently justified assumptions when summing the different ice and water layers collected from each pond. It is unclear what you mean by the 5% uncertainty in relation to the headspace method and storage in the water column, please provide a clear explanation and justification. Similarly, the arguments for including uncertainties related to peat are not clearly presented and require clarification.

We thank the reviewer for this comment! To provide more clarity, we have completely rearranged the Methods section on flux calculation, creating two subsections Sect. 3.5 "*$CH_4$ storage in water and ice*" and Sect. 3.6 "*Winter $CH_4$ bottom flux.*" In Sect. 3.5 we added the equations used to calculate the $CH_4$ winter storage (Eq. 1, revised manuscript) and the $CH_4$ storage prior to freezing (Eq. 2, revised manuscript). Based on these equations, we clarify how the uncertainty of each individual term in Eqs. 1 and 2 is estimated and how these uncertainties combine to a final uncertainty using Gaussian error propagation.

We state and justify a 5% uncertainty for $CH_4$ concentrations measured by the headspace method. This value is based on published error estimates for dissolved gases at low pH (Koschorreck et al., 2021); we assume the relative error for $CH_4$ is the same or lower than that reported for $CO_2$ under comparable conditions. We also clarify the treatment of uncertainties associated with frozen peat (Sect. 3.5, revised manuscript*)*: "*For the frozen peat samples, when only a single sample was available, we applied the average relative uncertainty from the deepest ice layers in other ponds.*"

In Sect. 3.6 of the revised manuscript, we again provide the defining equation (Eq. 3) for calculating the winter $CH_4$ bottom flux from the storage terms and describe the uncertainties associated with each of the terms. We then calculate the uncertainty of the winter $CH_4$ bottom flux using Gaussian error propagation.

Several methodological sections highlight potential problems with your core sampling procedure and the way the overall balance was calculated. Typically, storage estimates begin from the onset of ice cover and are calculated forward through the ice-covered period. In your study, however, you assume the end of the 2023–2024 ice period based on measurements from the beginning of the 2024–2025 ice period. This reversed logic is highly questionable. Please justify why the study design started in the opposite direction (you acknowledge it in the discussion but still is not enough

to consider a good selection, expand it and use literature to discuss about it). And would recommend to sort it properly, in Figure 2 or Table 2 you are sorting in a way that March measurements are later, which is not the case.

To derive the winter $CH_4$ bottom flux for the 2023–2024 season, we need to estimate the $CH_4$ storage in the pond water just before the onset of freezing in fall 2023. However, these measurements are unfortunately not available to us, as it is logistically very challenging to be out in the field exactly at the time of freeze-up for each of the ponds. For this reason, we used the $CH_4$ concentrations in the very first winter ice layer, as sampled in March 2024, which provides a fingerprint of the composition of dissolved gases at the time of freeze-up when the pond becomes decoupled from the atmosphere. However, we acknowledge that this approach introduces uncertainty, as it is not clear how fast the ice thickened and whether our "first winter ice" sample does not also contain layers with e.g. ebullition bubbles released after freeze-up. To cross-check the obtained values, we returned in September 2024 and re-sampled the ponds to obtain typical $CH_4$ concentrations during fall. After adjusting the dissolved $CH_4$ concentrations to 0 °C, the comparison confirms (Fig. S10) that, in most cases, the differences between the first-formed-ice concentrations and the September concentrations are relatively small, which suggests that they indeed provide an adequate estimate for the $CH_4$ concentrations prior to freezing. However, in a few cases the differences were much larger which we account for in our uncertainty analysis (see Sect.3.6, Sect. 4.3. Sect 5.1 in the revised manuscript). In case of larger differences, we assume very large uncertainties of up to 100% for the pre-freeze $CH_4$ storage, but since the absolute values are small compared to the winter storage, these uncertainties do not strongly contribute to the final uncertainty in the winter bottom flux. In the revised version, we have clarified this procedure in both the Methods (Sect. 3.6, revised manuscript) and the Discussion (Sect. 5.1, revised manuscript).

Concerning Fig. 2 and Table 2, we consider it most important to show the order of magnitude of the $CH_4$ increase from September to March rather than the strict chronological order of sampling. Instead, we clarify this in the figure caption: *"Note that the months are presented in non-chronological order to reflect the logical sequence of the winter CH4 accumulation."*

Regarding the sampling campaigns of dissolved gas in water and ice cores measurements, I have several questions. Because, measurements were very limited at the beginning of ice cover in October 2024, and those from September 2024 appear very superficial. Please clarify why dissolved gas samples were collected at only 0.1 m depth in September, and were sampling was conducted (in the center?) of the pond, and why not bottom samples were collected? Please explain the rationale for being selective in October 2024, why were some ponds sampled while others were not? The table showing pond properties is questionable not sampling them, as not all sites were included in the final sampling. Finally, how many samples were collected per site, only one ice core per pond? And water samples per point?

Due to logistical constraints (not all ponds were reachable due to thin ice) and poor weather, we could not sample all ponds in October 2024. We present the October data to show how quickly dissolved $CH_4$ storage increases with formation of the first ice on the ponds. We consider these data useful to the research community.

In September, we collected water samples at 0.1 m depth because the ponds are shallow and well mixed prior to first-ice formation, as supported by measured temperature profiles (Fig. S4). Samples are taken at the pond center, which we specified in the main text: *"The exact positions were determined from the aerial imagery (Sect. 3.7), generally in the central area of each pond, and located by differential GPS in the field."* (Sect. 3.1, revised manuscript).

We clarify the number of samples in Sect. 3.1 of the revised manuscript: one ice core per pond was collected. Generally, we split each core into three subsamples and used these subsamples for analysis which allowed us to evaluate the uncertainty in the $CH_4$ content of the ice. Water samples were taken from the same location where the ice core was taken. In winter, this was done at exactly the coring location after drilling a first hole through the ice. In this case, we took two samples where the remaining, unfrozen water column was deep enough, and only one sample when only a thin water layer was present. In September, the samples were taken from the shore at the approximate location of the winter coring in the center of the pond. Since the ponds are generally small and the water column is well-mixed during ice-free conditions (as suggested by the temperature measurements, see above), we consider this an adequate procedure.

Another critical part is the sampling procedure for dissolved gas and DOC measurements which also requires clarification. For example: (i) How much vacuum was created in the 12 mL vials prior to filling? Why was shaking performed for 5 minutes? This seems excessive, and the friction and hand-warm inside the syringe could have increased the temperature, thereby affecting gas solubility and Henry's law values. (ii) If acid was added directly into disposable syringes, this could have damaged the syringes and caused leaks. Were syringes replaced for each measurement? Did you check for potential sample interferences or leaks? If not, I strongly recommend verifying this in the laboratory. (iii)

We used a well-established sampling protocol for the dissolved gases, which has been used in our research group for many years, and which is documented in peer-reviewed articles, e.g.

1. Knutson, J. K., Clayer, F., Dörsch, P., Westermann, S., de Wit, H. A. Water chemistry and greenhouse gas concentrations in waterbodies of a thawing permafrost peatland complex in northern Norway, Biogeosciences, 22, 3899–3914, https://doi.org/10.5194/bg-22-3899-2025, 2025.
2. Eiler, A., Valiente Parra, N., Andersen, T., Hessen, D. O., Allesson, L. Drivers and variability of $CO_2$: O2 saturation along a gradient from boreal to Arctic lakes. Scientific Reports, 12(1), 18989–10, https://doi.org/10.1038/s41598-022-23705-9, 2022.

3. Wei, J., Fontaine, L., Valiente, N., Dörsch, P., Hessen, D. O., Eiler, A. Trajectories of freshwater microbial genomics and greenhouse gas saturation upon glacial retreat. Nature Communications, 14(1), 3234–12, https://doi.org/10.1038/s41467-023-38806-w, 2023.

Before applying the method in the field, we tested the equipment and procedures to rule out potential leaks, interferences, and sample damage. In particular, we followed the protocol described by Knutson et al. (2025) (who quantified summer gas concentrations and water chemistry at the Iškoras site) to maintain methodological consistency between different studies. In the revised version, we clarify the procedure in Sect. 3.1: *"Immediately after bringing the water samples to the surface, dissolved gases were extracted from a subsample on-site using the acidified headspace method (Åberg and Wallin, 2014) following the protocol of Knutson et al., (2025). 30 mL of water was collected into a 60 ml disposable syringe equipped with a 3-way valve and 20 mL headspace with ambient air was created. The samples were acidified with 0.6 mL of 3 % HCl to achieve a pH < 2, so that the dissolved inorganic carbon (DIC) was completely released as $CO_2$ into the headspace. To reach the equilibrium, the syringe was shaken for 1 min, followed by a 30 s rest and this sequence was repeated three times (Knutson et al., 2025). The headspace gas was transferred to a Helium (He) washed and evacuated 12 mL septum vials (Chromacol, remaining pressure 4-6 mbar)."*

We do not think that heating of the sample during equilibration was a major concern in our case, as the main sampling in March 2024 was conducted at freezing temperatures. Furthermore, we made sure to not touch the syringe with a warm hand (at all sampling dates), and we hold the plunger while shaking. The syringe is not insulated, so any frictional heating is likely to dissipate rapidly to the colder environment and not result in a major temperature change. Furthermore, the $CH_4$ solubility changes by only ~3% per 1°C, so that a small degree of warming could even be tolerated as it is negligible compared to other sources of uncertainty in our analysis.

Please note that Falcon tubes are known to leach DOC. Did you test whether this influenced your results? Filtration through 0.45 µm is unlikely to remove all bacteria, which could result in DOC depletion if samples were stored for too long. How long were DOC samples stored prior to analysis? (iv)

Using Falcon tubes and 0.45 µm filtration for DOC analysis is a standard practice in many published studies (e.g. Feng et al., 2020, Carlsen et al., 2025, Racasa et al., 2026). In addition, we tested for DOC leaching from the Falcon tubes and did not find any. For this purpose, we compared the results from lake samples stored in Falcon tubes with those stored in glass tubes, as well as with controls with deionized water; these comparisons did not show any detectable leaching. We filter samples through 0.45 µm filters, which remove most bacteria, and we store samples cold or frozen to minimize microbial activity. In our protocol, unfrozen samples are kept dark at 4 °C for no more than 7 days. The combination of filtering and short-time storage at cold temperatures inhibits bacterial growth or alteration of DOC. In the revised manuscript, we added the storage conditions and maximum storage time to the Methods (Sect. 3.1, revised manuscript).

I do not consider your reported $CO_2$ values from dissolved gas samples to be valid, since total inorganic carbon was not determined. Without this measurement, the reported $CO_2$ concentrations cannot be considered representative of the actual conditions in the water (you added acid and no alkalinity was measured), or you need to expand the calculation of Appelo and Postma, 1993. (v) the type of GC detector is not clear, and also you must provide the detection limit for the gases. Also, I do not see the point to include $CO_2$ and N2O in the study since the study is focused on $CH_4$ (N2O is mentioned in the methodology but not used in the results or discussion). (vi) the O2 is not clear how did you measure and which device was used for it. (vii) Again the mixing of the ice samples in the jars was for 1 hour to equilibrate headspace, the remaining oxygen in the ice could be used to oxidize the methane stored in the ice. Still I do not understand why you have such long periods of mixing.

We agree with the reviewer that $CO_2$ concentrations and $CO_2$:$CH_4$ ratios were not the main focus of our study. However, we believe these data can provide potentially important context for pond classification and serve as future reference, e.g. in modeling studies on pond greenhouse gas balances. Therefore, we have moved the dissolved $CO_2$ results to a new Section 2.1 in the Supplementary Materials. In the Results section of the main paper, we only provide a short reference to Supplementary Section 2.1.

We thank the reviewer for the comment on the inorganic carbon which helped us correct the $CO_2$ values for ponds with pH > 5. We use the acidified headspace method of Åberg and Wallin (2014) for inorganic carbon. After acidification the sample pH is 2 and dissolved inorganic carbon (DIC) is released completely as $CO_2$ into the headspace; we now state this clearly in the Methods section of the revised manuscript (Sect. 3.3). We also corrected the dissolved $CO_2$ values calculated from total DIC after acidification, using in situ pH and equilibrium constants adjusted to pond temperature following Eq. 12 in Åberg and Wallin (2014). When the pH is below 5, this correction is negligible. However, for the two ponds with pH > 5 (A3 and A8) we recalculated dissolved $CO_2$ and include those values in the revised Supplementary Section 2.1 (Fig. S9a, revised supplementary). The recalculated $CO_2$ concentrations for these two ponds are 1.1–1.7 times smaller (a decrease of 9–42%) than the uncorrected values. As Fig. S9a uses a logarithmic scale, these changes are hardly visible, and the main conclusion from the figure remains unchanged.

In September, we measured $O_2$ together with the other gases to characterize conditions prior to freezing; this is now reported in the Methods (Sect. 3.3, revised manuscript). Furthermore, we now specify the GC setup in the Methods (Sect. 3.3, revised manuscript): "*CH$_4$ was measured with a flame-ionization detector (FID; detection limit 0.1 ppm). CO$_2$, O2 and N2 were measured with a thermal-conductivity detector (TCD; detection limits 10 ppm for CO$_2$ and 100 ppm for O2 and N2)."* We removed N$_2$O as an operational characteristic of the gas chromatograph in Sect. 3.3.

To avoid oxidation during headspace extraction, we flushed the ice samples with He prior to melting. We reviewed the relevant literature on methane oxidation rates to evaluate potential impacts on the resulting $CH_4$ concentrations. Reported oxidation rates in similar subarctic and

boreal surface waters are in the range 0.0007–0.05 µmol $CH_4$ $L^{-1}$ hour$^{-1}$ (Matveev et al., 2018; Kankaala et al., 2006). Our measured $CH_4$ concentrations in the ice samples vary from 0.1 to 1258 µmol $CH_4$ $L^{-1}$ w.e. Assuming the maximum reported $CH_4$ oxidation rate (0.05 µmol $L^{-1}$ $h^{-1}$), oxidation during 1-hour equilibration would correspond to 50% of our smallest measured concentration (0.1 µmol $L^{-1}$) and $\leq 0.004\%$ of our maximum measured concentration (1258 µmol $L^{-1}$). For the vast majority of samples, this error source is negligible. In addition, the lowest concentrations (typically in the first-formed ice layer where the relative error of oxidation would be highest) contribute very little to the total ice $CH_4$ storage. For this reason, we do not think that oxidation during equilibration can meaningfully alter $CH_4$ concentrations and thus affect the calculated ice storage and winter bottom-flux estimates.

Figure 2 are showing some error bars, what is this and how they were estimated, it is not clear in the text. Please sort it properly, March at the beginning.

Regarding Figure 2 and Table 2, as explained above, we consider it most important to show the strong $CH_4$ increase from September to March, rather than the exact chronological order. However, we clearly state this now in in the figure caption: *"Note that the months are presented in non-chronological order to reflect the logical sequence of the winter $CH_4$ accumulation."* Error bars represent standard deviations of multiple samples (n = 3–12) collected from different depths in March and October, as well as replicate samples taken in September from the same depth. We have specified this in the capture of Figure 2.

Figure 3 is a boxplot, so please add the number of data used to construct them, and the meaning of the whiskers, boxes and lines and circles.

We thank the reviewer for the comment. We have corrected Figure 3 and revised the caption to specify the number of data points used. The revised caption now reads: *"Figure 3. Box plots illustrating methane ($CH_4$) concentrations in distinct ice types (ice types from Boereboom et al., 2012 with adjustments): 1 – Superimposed ice (n = 20), 2 – Clear ice (n = 31), 3 – Methane ebullition bubbles (n = 9), 4 – Spherical and nut–shaped bubbles (n = 10), 5 – Elongated bubbles (12), 6 – Mixed bubbles (n=38). Boxes show the interquartile range (25th–75th percentiles), the line indicates the mean, and whiskers extend to the min and max values. For Frozen peat (7) with n = 2, only mean, min and max are shown. $CH_4$ concentrations are reported on a water-equivalent (w.e.) basis".*

The discussion and conclusion sections are highly repetitive. I recommend condensing them and reformulating after the methodology and results have been corrected or modified in response to my previous comments.

We agree with the reviewer that especially the Discussion section needed to be streamlined and shortened. We have completely rewritten the Methods section for clarity and revised Results and Discussion accordingly. Furthermore, in response to the reviewer's comment, we shortened the Discussion, removing some of the less focused discussion points from Sections 5.2 and 5.3.

In addition, Figure 6 is not sufficiently supported by the results and appears to present data in a casual way. Please rework this figure to ensure that it is consistent with, and properly supported by, your findings.

Fig. 6 shows the winter $CH_4$ bottom fluxes (i.e. the same values as in Fig. 5) for the main study area in Áidejávri, plotted against the formation age of the thermokarst ponds. Furthermore, we provide an assessment of the succession stage of the ponds in the figure. In response to the reviewer's comment, we have revised the Methods section to add clear information on how the two latter quantities (age and succession stage) were obtained (Section 3.7, revised manuscript). Furthermore, we added horizontal error bars to indicate the uncertainty in the timing of thermokarst pond formation, based on historical and drone aerial imagery (as detailed in Sect. 3.7, revised manuscript).

References:

Bastviken, D., Johnson, M.S. Future methane emissions from lakes and reservoirs. Nature Water, 3, 1397–1410. https://doi.org/10.1038/s44221-025-00532-6, 2025.

Boereboom, T., Depoorter, M., Coppens, S., Tison, J.-L. Gas properties of winter lake ice in Northern Sweden: implication for carbon gas release, Biogeosciences, 9, 827–838, https://doi.org/10.5194/bg-9-827-2012, 2012.

Carlsen, E. C. L., Wei, J., Lejzerowicz, F., Trier Kjær, S., Westermann, S., Hessen, D. O., Dörsch, P., Eiler, A. Redox determines greenhouse gas production kinetics and metabolic traits in water-saturated thawing permafrost peat. ISME Communications, 5(1), ycaf009. https://doi.org/10.1093/ismeco/ycaf009, 2025.

Feng, L., An, Y., Xu, J., Li, X., Jiang, B., Liao, Y. Biochemical evolution of dissolved organic matter during snow metamorphism across the ablation season for a glacier on the central Tibetan Plateau. Scientific Reports,10 (1), 6123. https://doi.org/10.1038/s41598-020-62851-w, 2020.

Kankaala, P., Huotari, J., Peltomaa, E., Saloranta, T., Ojala, A. Methanotrophic activity in relation to methane efflux and total heterotrophic bacterial production in a stratified, humic, boreal lake. Limnology and Oceanography, 51(2), 1195–1204. https://doi.org/10.4319/lo.2006.51.2.1195, 2006

Knutson, J. K., Clayer, F., Dörsch, P., Westermann, S., de Wit, H. A. Water chemistry and greenhouse gas concentrations in waterbodies of a thawing permafrost peatland complex in northern Norway, Biogeosciences, 22, 3899–3914, https://doi.org/10.5194/bg-22-3899-2025, 2025.

Koschorreck, M., Prairie, Y. T., Kim, J., Marcé, R. Technical note: $CO_2$ is not like $CH_4$ – limits of and corrections to the headspace method to analyse p $CO_2$ in fresh water, Biogeosciences, 18, 1619–1627, https://doi.org/10.5194/bg-18-1619-2021, 2021.

Kuhn, M. A., Varner, R. K., Bastviken, D., Crill, P., MacIntyre, S., Turetsky, M., Walter Anthony, K., McGuire, A. D., and Olefeldt, D. BAWLD-$CH_4$: a comprehensive dataset of methane fluxes from boreal and arctic ecosystems, Earth System Science Data, 13, 5151–5189, https://doi.org/10.5194/essd-13-5151-2021, 2021.

Matveev, A., Laurion, I., Vincent, W. F. Methane and carbon dioxide emissions from thermokarst lakes on mineral soils. Arctic Science, 4(4), 584–604. https://doi.org/10.1139/as-2017-0047, 2018

Pirk, N., Aalstad, K., Mannerfelt, E. S., Clayer, F., De Wit, H., Christiansen, C. T., Althuizen, I., Lee, H., Westermann, S. Disaggregating the Carbon Exchange of Degrading Permafrost Peatlands Using Bayesian Deep Learning, Geophysical Research Letters, 51, e2024GL109283, https://doi.org/10.1029/2024GL109283, 2024.

Racasa, E. D., Jenner, A.-K., Saban, R., Kienzler, J., Batistel, C., Choo, S., Wang, M., Gräwe, U., Böttcher, M. E., Janssen, M. Characterization of Submarine Groundwater Discharge in Front of a Rewetted Coastal Peatland. Estuaries and Coasts, 49(1), 12. https://doi.org/10.1007/s12237-025-01607-z, 2026

Vonk, J. E., Tank, S. E., Bowden, W. B., Laurion, I., Vincent, W. F., Alekseychik, P., Amyot, M., Billet, M. F., Canário, J., Cory, R. M., Deshpande, B. N., Helbig, M., Jammet, M., Karlsson, J., Larouche, J., MacMillan, G., Rautio, M., Walter Anthony, K. M., Wickland, K. P. Reviews and syntheses: Effects of permafrost thaw on Arctic aquatic ecosystems, Biogeosciences, 12, 7129–7167, https://doi.org/10.5194/bg-12-7129-2015, 2015.